# Interaction between Lifestyle Changes and PNPLA3 Genotype in NAFLD Patients during the COVID-19 Lockdown

**DOI:** 10.3390/nu14030556

**Published:** 2022-01-27

**Authors:** Felice Cinque, Annalisa Cespiati, Rosa Lombardi, Andrea Costantino, Gabriele Maffi, Francesca Alletto, Lucia Colavolpe, Paolo Francione, Giovanna Oberti, Erika Fatta, Cristina Bertelli, Giordano Sigon, Paola Dongiovanni, Maurizio Vecchi, Silvia Fargion, Anna Ludovica Fracanzani

**Affiliations:** 1Unit of Internal Medicine and Metabolic Disease, Fondazione Ca’ Granda IRCCS Ospedale Maggiore Policlinico, 20122 Milan, Italy; felice.cinque@unimi.it (F.C.); annalisa.cespiati@unimi.it (A.C.); gabriele.maffi1@unimi.it (G.M.); francesca.alletto@unimi.it (F.A.); lucia.colavolpe@unimi.it (L.C.); paolo.francione@unimi.it (P.F.); giovanna.oberti@unimi.it (G.O.); erika.fatta@policlinico.mi.it (E.F.); cristina.bertelli@policlinico.mi.it (C.B.); giordano.sigon@unimi.it (G.S.); paola.dongiovanni@policlinico.mi.it (P.D.); silvia.fargion@unimi.it (S.F.); anna.fracanzani@unimi.it (A.L.F.); 2Department of Pathophysiology and Transplantation, University of Milan, 20122 Milan, Italy; andrea.costantino@unimi.it (A.C.); maurizio.vecchi@unimi.it (M.V.); 3Unit of Gastroenterology and Endoscopy, Fondazione Ca’ Granda IRCCS Ospedale Maggiore Policlinico, 20122 Milan, Italy

**Keywords:** NAFLD, Mediterranean diet, physical activity, PNPLA3, weight gain, lockdown, SARS-CoV-2

## Abstract

The coronavirus disease 2019 (COVID-19) lockdown dramatically changed people’s lifestyles. Diet, physical activity, and the PNPLA3 gene are known risk factors for non-alcoholic fatty liver disease (NAFLD). Aim: To evaluate changes in metabolic and hepatic disease in NAFLD patients after the COVID-19 lockdown. Three hundred and fifty seven NAFLD patients were enrolled, all previously instructed to follow a Mediterranean diet (MD). Anthropometric, metabolic, and laboratory data were collected before the COVID-19 lockdown in Italy and 6 months apart, along with ultrasound (US) steatosis grading and information about adherence to MD and physical activity (PA). In 188 patients, PNPLA3 genotyping was performed. After the lockdown, 48% of patients gained weight, while 16% had a worsened steatosis grade. Weight gain was associated with poor adherence to MD (*p* = 0.005), reduced PA (*p* = 0.03), and increased prevalence of PNPLA3 GG (*p* = 0.04). At multivariate analysis (corrected for age, sex, MD, PA, and PNPLA3 GG), only PNPLA3 remained independently associated with weight gain (*p* = 0.04), which was also associated with worsened glycemia (*p* = 0.002) and transaminases (*p* = 0.02). During lockdown, due to a dramatic change in lifestyles, half of our cohort of NAFLD patients gained weight, with a worsening of metabolic and hepatologic features. Interestingly, the PNPLA3 GG genotype nullified the effect of lifestyle and emerged as an independent risk factor for weight gain, opening new perspectives in NAFLD patient care.

## 1. Introduction

Nonalcoholic fatty liver disease (NAFLD) is defined by the presence of fat in more than 5% of hepatocytes in the absence of other causes of liver disease [1]. The term NAFLD encloses a spectrum of different conditions, ranging from simple steatosis to steatohepatitis (NASH) where inflammation coexists, possibly evolving to fibrosis and cirrhosis. Metabolic alterations, mostly type 2 diabetes (T2DM) and obesity [2,3], as well as genetic predisposition, especially polymorphism in the patatin-like phospholipase domain-containing protein 3 (PNPLA3) gene [4,5,6,7], are risk factors for NAFLD onset and progression towards advanced liver disease.

The first line and most effective treatment for both metabolic syndrome (MeTS) and NAFLD is weight loss, achieved by diet and physical exercise [8,9]. Weight loss, especially if >5–10% from baseline, promotes improvement in hepatic steatosis, inflammation, and fibrosis [10]. On the other hand, weight gain increases the risk of advanced liver disease and hepatocellular carcinoma (HCC) [9,11]. The most recommended diet regimen in patients with NAFLD is the Mediterranean diet (MD) [12,13,14,15,16], characterized by reduced daily intake of carbohydrates (30% of the whole daily caloric intake, preferably whole grain) and high fat consumption (35–45% of the total energy intake), mainly mono-unsaturated fatty acids (MUFAs) and poly-unsaturated fatty acids (PUFAs)—coming from olive oil, nuts, and fish—with a low intake of saturated fatty acids (<8% of total daily calories) [17]. Sugary drinks and high fructose corn syrup (HFCS) content foods are forbidden; conversely, increased amounts of fibers found in vegetables, legumes, and whole grains are recommended [18]. A small amount of alcohol consumption, below 30 g/day in men and 20 g/day in women, is permitted [9]. Regarding physical activity (PA), patients with NAFLD are encouraged to perform at least 150 min/week of moderate intensity physical activity over three to five sessions, combining both aerobic and resistance training [19,20,21,22,23].

In December 2019, a viral infection sustained by a novel coronavirus family, named Severe Acute Respiratory Syndrome Coronavirus-2 (SARS-CoV-2) and called COVID-19, was described in China [24] and subsequently spread across the world, with the World Health Organization (WHO) declaring it a pandemic in March 2020. Because of a high rate of morbidity and mortality, many countries imposed quarantine policies in order to control the spread of the infection [25]. Italy imposed a nation-wide lockdown from March to April 2020.

If, on the one hand, these measures were highly effective in containing the spread of the SARS-CoV-2 infection, on the other hand, they limited outdoor activities and promoted smart working at home, with consequent changes in social and dietary habits [26]. As a consequence of the lockdown, weight gain in the general population [27,28] has been described, as well as deterioration in glycemic control in patients with T2DM [29,30,31].

Given the close association between NAFLD, weight gain, and metabolic alterations, and the lack of data in the literature on the impact of the lockdown on NAFLD patients, the aim of our study was to evaluate changes in hepatic and metabolic features in a cohort of Italian NAFLD patients during the COVID-19 pandemic. We also investigated the impact of the PNPLA3 genotype on the effect of lifestyle changes in the same cohort.

## 2. Materials and Methods

### 2.1. Patients

We performed a retrospective study in a cohort of 357 patients (mean age was 61 ± 12 years, 67% male) with a diagnosis of NAFLD, according to the European Study of Liver Disease (EASL) criteria [9], who were referred to the Metabolic and Liver Disease outpatient clinic of the Fondazione Ca’ Granda IRCCS Ospedale Maggiore Policlinico, in Milan, Italy. Enrolled patients had one medical checkup between June and October 2020, and all previous checkups had been performed no more than 6 months before the COVID-19 lockdown. For all patients, anthropometric, clinical, and biochemical data and assessment of lifestyle were collected at the two visits. Patients with HCC or decompensated cirrhosis were excluded from the study. None of the patients had had a COVID-19 infection that required hospitalization. The study protocol was approved by the Institutional Review Board. For all patients, informed consent to participate to the study was obtained according to the ethical guidelines of the 1975 Declaration of Helsinki.

### 2.2. Assessment of Dietary Habits and Lifestyle

As for our routine clinical practice, assessment of adherence to the MD and PA, along with smoking habits and alcohol consumption, were performed at each follow-up visit, so this information was collected before and after the COVID-19 lockdown. In particular, at all visits, patients’ dietary habits were compared to a balanced MD: patients were considered adherent to the MD if they consumed low carbohydrates (30% of the whole daily calories intake, preferably whole grain) and low saturated fat (<8% of total energy intake), while consuming a high proportion of mono-unsaturated fatty acids and polyunsaturated acid (30–40% of energy intake), with no consumption of sugary drinks nor high fructose corn syrup (HFCS) content foods. PA was evaluated by asking patients the type and amount of PA they performed. Patients were classified in the aerobic PA group if they performed an aerobic PA of at least 150–300 min of moderate intensity or of 75–150 min a week of vigorous intensity; conversely, they were included in the resistance PA if they performed a series of 8–10 whole-body exercises (exercises involving each major muscle group) at least twice a week for 60 min during each session [32]. Patients performing either aerobic PA or resistance PA were classified as physically active, otherwise they were considered inactive.

### 2.3. Assessment of Metabolic Comorbidities

Normal weight, overweight, and obesity were defined by the body mass index (BMI) values of <25, 25–29, and ≥30 kg/m^2^. Patients presenting with in-office blood pressure values ≥140/90 mmHg or who used any antihypertensive drugs [33] were considered hypertensive. Dyslipidemia was defined by low-density lipoprotein (LDL) cholesterol >100 mg/dL and/or triglycerides >150 mg/dL and/or high-density lipoprotein (HDL) cholesterol <40 mg/dL for men and 50 mg/dL for women and/or use of lipid-lowering drugs [34]. T2DM was diagnosed by the presence of fasting glucose >126 mg/dL in more than two consecutive measurements or a single value >200 mg/dL or a glycated hemoglobin (Hb1Ac) >6.5% (48 mmol/mol) or therapy with any hypoglycemic drugs [35], as well as retrieved by medical history. Blood samples were obtained after a fasting night. The biochemical panel included a glucose-and-lipids profile.

### 2.4. Assessment of Liver Disease

According to local laboratory cut-offs, aspartate aminotransferase (AST) ≥39 U/L, alanine aminotransferase (ALT) ≥41 U/L and gamma-glutamyltransferase (GGT) ≥61 UI for men and >36 UI/L for women were considered increased values. The presence and grade of hepatic steatosis was evaluated by ultrasound (US), according to the literature [36,37,38]. The presence and severity of hepatic fibrosis were detected by non-invasive fibrosis score (Fibrosis-4 (FIB4) score). Cut-offs of 1.3 and 2.67 were used to rule-out and rule-in fibrosis, respectively [39]. In a subset of 80 patients, a previous liver biopsy was available.

### 2.5. Genetic Analysis

Data about the rs738409 C > G (I148M PNPLA3) single nucleotide polymorphism by TaqMan 5′-nuclease assays (Life Technologies, Carlsbad, CA, USA) were available in 188 patients.

### 2.6. Statistical Analysis

Continuous variables were expressed as means ± SD or medians (interquartile range) for normally and not normally distributed variables, respectively. Categorical variables were expressed as absolute and relative frequencies (*n*, %). In order to compare differences between groups, we used the chi-squared test for categorical variables and the unpaired Student’s t-test or the Wilcoxon test for continuous ones, according to data distribution. To evaluate changes in metabolic and lifestyle variables before and after lockdown, a paired t-test and the independent sample proportions were used for continuous and categorical variables. A multivariable logistic regression analyses was performed to evaluate factors associated to increased weight after lockdown, adjusted for confounding factors (age, sex, MD, and PA during lock-down and PNPLA3 genotype) and tested, assuming an additive model. A two-tailed *p*-value ≤ 0.05 was considered statistically significant. Statistical analyses were performed using JMP 15.2 (SAS Institute, Cary, NC, USA).

## 3. Results

### 3.1. Anthropometric and Lifestyle Characteristics at Pre-Lockdown Visit

Thirty-nine (11%) patients were current smokers, and 23% used to drink small amounts of alcohol (median alcohol intake 12 (6–18) g/day).

Before the lockdown, in the entire cohort, 186 (56%) patients referred to following a balanced MD, while 128 (37%) referred that they performed regular physical activity (35% aerobic PA and 2% resistance PA).

### 3.2. Metabolic and Hepatic Features at Pre-Lockdown Visit

The mean body weight was 80 ± 14 Kg, with 52% of patients overweight and 30% obese.

The prevalence of hypertension, T2DM (mean glycemia 106 ± 23 mg/dL), and dyslipidemia (mean LDL 102 ± 33 mg/dL, HDL 50 ± 13 mg/dL; median triglycerides 118 (IQR 90-163) mg/dL) were 55%, 23%, and 53%, respectively.

Increased ALT were observed in 24%, AST in 9%, and GGT in 29%. Grading of steatosis at US showed grade 1 in 44%, grade 2 in 41%, and grade 3 in 15% of patients, whereas advanced fibrosis by non-invasive score FIB-4 was diagnosed in 5% of and ruled out in 51% patients. In the subset of patients with an available liver biopsy (*n* = 80), NASH was present in 70 (87%) cases, of whom mild fibrosis (F1) was present in 27 (38%), significant fibrosis (F2) in 17 (24%), advanced fibrosis (F3) in 12 (17%), and cirrhosis in 2 (3%).

### 3.3. Changes in Hepatic and Metabolic Features after Lockdown

After the COVID-19 lockdown, 48% of the cohort (171 patients) had an increase in body weight, with a mean weight gain of 3.2 ± 4 Kg (29% of the cohort with weight gain >5% and 4% >10%). Conversely, in 17% of patients, body weight remained stable, whereas 35% of the cohort lost weight, with a mean weight loss of 2.7 ± 2.7 Kg.

Similarly, in 57 (16%) patients, an increase in steatosis US grade was registered at the post-lockdown visit, without any progression of fibrosis by FIB-4 in the whole cohort (data not shown).

Finally, no change in the prevalence of metabolic alterations (T2DM, hypertension, and dyslipidemia) was observed before and after lockdown.

### 3.4. Factors Determining Weight Gain in the Whole Cohort

As shown in Table 1, during lockdown, patients who gained weight had a significantly reduced adherence to a balanced MD (*p* = 0.005) and performed reduced PA (*p* = 0.03) compared to those who did not.

Conversely, no difference in age, sex, prevalence of metabolic comorbidities, and smoking and alcohol habits was found.

Similarly, the severity of liver disease at the baseline did not influence weight gain. Interestingly, patients who increased in body weight had the GG PNPLA3 genotype more frequently than patients who did not (24% vs. 12% *p* = 0.04).

In order to differentiate the impact on the weight gain of lifestyle changes and genetic factors—alone (Table 2, model 1, 2, 3) or in combination (Table 2, model 4, 5)—a multivariate analysis corrected for age and sex was performed. When analyzed alone, both lifestyle changes (adherence to MD, OR 0.52, CI 95% 0.34–0.82 *p* = 0.004; PA, OR 0.6, CI 95% 0.37–0.95 *p* = 0.03) and genetic factors (PNPLA3 homozygosity for the G allele, OR 2.4, CI 95% 1.08–5.32, *p* = 0.03) resulted as independent risk factors for weight gain. Analyzing adherence to MD and PA together (Table 2, model 4), both variables (OR 0.54, CI 95% 0.34–0.87 *p* = 0.01; OR 0.58, CI 95% 0.35–0.96 *p* = 0.03) were independently associated with weight gain. When considering the coexistence of the three variables—MD adherence, PA, and PNPLA3GG—(Table 2, model 5) only the PNPLA3 GG genotype (OR 2.391, CI 95% 1.02–5.58; *p* = 0.04) remained significantly associated with weight gain.

BMI, body mass index. T2DM, type 2 diabetes. US, ultrasound. FIB-4, fibrosis 4 score. NASH, non-alcoholic steatohepatitis. MD, Mediterranean diet. PA, physical activity.

### 3.5. Impact of Lockdown on Lifestyle, Metabolic, and Hepatic Features in the Entire Cohort According to Weight Gain

As shown in Table 3, adherence to MD remained stable and low in the whole cohort before and after lockdown, irrespective of the body weight change. Conversely, a lower adherence to PA was significantly evident in the subset of patients who gained weight (35% vs. 25%, *p* = 0.02), without any modification in those who did not.

In addition, the prevalence of metabolic alterations (i.e., T2DM, hypertension and dyslipidemia) did not change before and after lockdown, irrespective of weight change.

Conversely, as for biochemical metabolic parameters, significantly higher glycemia levels were registered after lockdown only in patients who gained weight (112 ± 32 vs. 106 ± 25, *p* = 0.002), without any change in lipids.

On the contrary, no change in all biochemical parameters was observed in patients who maintained a stable weight during the lockdown period.

As for liver disease, a higher prevalence of steatosis grade ≥2 after the lockdown was registered in patients who gained weight (73% vs. 59%, *p* = 0.002) but not in those who did not (56% vs. 52%, *p* = 0.21), without any worsening of fibrosis by FIB-4 in both groups. In addition, a higher prevalence of increased transaminases and GGT was observed after the lockdown only among those who gained weight (ALT 30% vs. 21% *p* = 0.02; GGT 35% vs. 25%, *p* = 0.01).

### 3.6. Role of Age

Given the change also in working habits during lockdown and considering the retirement age in Italy, which is approximately 67 years old, we divided our cohort in younger patients (222 subjects < 67 years old, 71% male) and older ones (135 subjects ≥ 67 years old, 62% male).

Differences between groups are described in Appendix A. In particular, before the lockdown, both groups had similar rates of adherence to MD and PA (56% vs. 54%, *p* = 1.00 and 38% vs. 35%, *p* = 0.57, respectively).

Analyzing modifications in lifestyle, metabolic, and hepatic features before and after the lockdown, according to age (i.e., < or >67 ys) and in patients who gained weight compared to those who did not (Appendix A), both younger and older subjects showed a reduced adherence to MD (patients with age <67 ys: 44% vs. 58%, *p* = 0.001, patients with age >67 ys: 49% vs. 65%, *p* = 0.03). Interestingly, younger patients who increased in weight had a significantly reduced PA during lockdown (23% vs. 40%, *p* = 0.002), while older patients did not show a reduction in PA (32% vs. 28%, *p* = 0.59).

No change in the prevalence of metabolic comorbidities was registered before and after lockdown, irrespective of age and weight change. Conversely, an increase in glycemia levels (109 ± 32 vs. 103 ± 23, *p* = 0.04) was observed after lockdown in younger patients who increased in weight but not in those who did not gain weight, while biochemical parameters remained stable in older subjects, irrespectively of weight change.

As for liver disease, hepatic features worsened only in the groups who gained weight. In particular, a higher prevalence of US hepatic steatosis grade ≥2 was observed after the lockdown in older patients (64% vs. 46%, *p* = 0.007), whereas a higher prevalence of increased liver enzyme was seen in younger patients (increased ALT 38% vs. 25%, *p* = 0.005, AST 15% vs. 9%, *p* = 0.05, GGT 38% vs. 28%, *p* = 0.02). No worsening of fibrosis by FIB-4 was observed across all groups.

## 4. Discussion

In this retrospective study, we demonstrated the negative impact of the COVID-19 lockdown on NAFLD because of worsening lifestyle habits, which play a key role in the therapy for these patients. In addition, we showed an independent association between the PNPLA3 GG genotype and weight gain.

Almost half of the cohort increased in weight, with up to 29% of patients gaining more than 5% of their basal weight. Worsening of steatosis was detected in 16% of the cohort, while no change in fibrosis was seen, probably because the time that elapsed between the two follow-up visits was too short. Indeed, modification of lifestyle was the main driver of weight gain, since both the reduction in physical activity (significantly reduced after lockdown) and a low adherence to the Mediterranean diet (around 50%) were independent risk factors for it. Worthy of note, the PNPLA3 GG genotype seemed to be the most important determinant of weight gain, and, interestingly, in patients carrying PNPLA3 GG allele, the impact of lifestyle was almost nullified, with the genetic polymorphism being the strongest predictor.

Lifestyle interventions, including a correct MD and regular PA with weight loss, represent the sole approved treatment for NAFLD [40,41,42,43,44,45,46,47], being strongly recommended in both European and American guidelines for NAFLD treatment [1,9]. Our results parallel those of other studies in which quarantine measures with consequent reduced PA and poor dietary habits led to an increase in body weight in the general population [27,28], obese patients [48], and diabetic outpatients [49,50]. One study including few patients reported weight gain in NAFLD subjects after lockdown [51], but results were based only on a self-administered questionnaire. Therefore, to our knowledge, this is the first study evaluating the impact of the COVID-19 lockdown in a large cohort of ultrasound-proven NAFLD patients, with an extensive evaluation of hepatic and metabolic alterations.

We demonstrated an independent association between the PNPLA3 GG genotype and weight gain, irrespective of unhealthy lifestyle habits. PNPLA3, expressed both in hepatocytes and adipocytes, has a role in lipid droplets storage [52]. PNPLA3 GG allelic expression is known for conferring an augmented risk of development and severity of NALFD [7,53,54] and seems to modify the impact of different therapeutic strategies for NAFLD [55]. In addition, PNPLA3 has been reported [52,56,57,58] to influence the accumulation of fat in adipose tissue and dietary changes—especially increased carbohydrates consumption [59]—may modulate its expression, possibly explaining the role of PNPLA3 in weight gain. Moreover, lower levels of adiponectin, an anti-inflammatory cytokine inversely associated with NAFLD, insulin-resistance, and impairment in glucose and lipid metabolism [60], have been found in both PNPLA3 GG subjects and obese patients, even without NAFLD [61]. Focusing on the impact of the PNPLA3 genotype on lifestyle interventions in NAFLD patients, one study showed that mutated patients were more sensitive to the beneficial effect of lifestyle modification on steatosis reduction, rather than wildtype patients [62]. Another study comparing the effect of a dietary intervention between subjects with PNPLA3 GG genotype and wildtype ones has demonstrated that mutated patients exhibited a greater weight loss and, as a consequence, a greater reduction in liver steatosis compared to not mutated ones [63]. Further studies that aim to confirm the role of PNPLA3 in weight gain and to compare weight change after physical activity in patients after lockdown according to PNPLA3 genotype are warranted.

Despite the fact that the mean weight gain was only 3 kg, a worsening of hepatic steatosis was demonstrated. In addition, high rates of reduced glycemic control and transaminases derangement were observed after lockdown in patients who increased their weight. Indeed, an impairment in glucose control during the COVID-19 pandemic was already reported in diabetic patients, and, similar to our data, it was caused by a change in diet and physical activity [29,30,31,64]. Conversely, no data are available in the literature on the worsening in transaminases after the COVID-19 lockdown, particularly in NAFLD patients. Therefore, it is important to stress that weight control in patients with NAFLD is essential in order to prevent worsening of the liver and metabolic control.

Finally, the present study shows that the lockdown had a different impact on patients’ lifestyle depending on age. In fact, in patients who gained weight, along with reduced adherence to MD of the whole cohort, the impact of reduced physical activity was mainly evident in younger patients (i.e., <67 years) compared to older ones. This could be possibly explained by the dramatic change in daily routines, which prevalently concerned younger patients, more likely to be active workers forced to smart working at home and with reduced chances of performing regular outdoor physical activity.

Our study has some limitations. First of all, the retrospective nature of the study prevents us from drawing a definite causative effect between change in lifestyle during lockdown and the worsening of metabolic and liver control, although similar results in different cohorts of patients have been reported. Moreover, regarding the assessment of lifestyle, no standardized quantitative questionnaires were administrated for the evaluation of dietary habits. However, accurate dietary counseling was provided to all patients, and, most importantly, adherence to MD was assessed at each visit. Finally, the PNPLA3 genetic characterization was available in a subset of patients (188 subjects), and, despite the need for further studies involving a larger cohort to further investigate the impact of PNPLA3 on weight gain, our data seem consistent with literature.

## 5. Conclusions

We demonstrated for the first time that the COVID-19 lockdown had a dramatic negative impact in the NAFLD population, leading to weight gain and the worsening of steatosis, consequent to a deterioration of healthy lifestyle habits. Most interestingly, a role for PNPLA3 emerged as a key independent risk factor for weight gain, opening new perspectives in the risk stratification and follow-up of NAFLD patients, especially in patients who could not regularly attend the hepatology clinic. Indeed, the COVID-19 pandemic possibly highlighted the need for reconsideration of the modality of assistance to NAFLD patients in the achievement of an appropriate lifestyle. In fact, the European Society for the Study of the Liver (EASL), in its position article on the care of hepatologic patients during the COVID-19 pandemic [65], has also underscored the need to continue to follow standard care of NAFLD outpatients during pandemic emergencies, possibly also using telemedicine, which is emerging as a good strategy to improve patient follow up [66,67].

## Figures and Tables

**Table 1 nutrients-14-00556-t001:** Univariate analysis of factors determining weight gain in the entire cohort.

Variable	Not Increased Weight (*N* = 186)	Increased Weight (*N* = 171)	*p* Value
Baseline Characteristics			
Sex, male, *n* (%)	130 (70)	108 (63)	0.21
Age, ys	62 ± 13	61 ± 11	0.29
Current smokers, *n* (%)	20 (11)	19 (11)	0.75
Light Drinkers, *n* (%) Alcohol intake, g/day	48 (26) 12 (6–22.5)	31 (18) 9 (6–12)	0.26 0.08
BMI, kg/m^2^ ➢ Normal weight ➢ Overweight ➢ Obese	28.3 ± 3.9 32 (17) 102 (55) 52 (28)	28.5 ± 4.7 33 (19) 82 (48) 56 (33)	0.77 0.41
Hypertension, *n* (%)	102 (55)	94 (55)	1.00
T2DM, *n* (%)	41 (22)	43 (25)	0.62
Dyslipidaemia, *n* (%)	97 (52)	92 (54)	0.91
US steatosis ➢ 1, *n* (%) ➢ 2, *n* (%) ➢ 3, *n* (%)	87 (47) 71 (39) 28 (15)	70 (41) 75 (44) 26 (15)	0.37
FIB-4 <1.3, *n* (%)	78 (42)	77 (45)	0.65
FIB-4 >2.67, *n* (%)	6 (3)	8 (5)	0.59
Presence of NASH ^1^, *n* (%)	31 (17)	39 (23)	0.11
PNPLA3 ^2^ ➢ GG, *n* (%) ➢ CG + CC, *n* (%)	11 (12) 81 (88)	23 (24) 73 (76)	0.04
Lifestyle During Lockdown			
Balanced MD, *n* (%)	100 (54)	72 (42)	0.005
Regular PA, *n* (%)	67 (36)	43 (25)	0.03

Analysis adjusted for sex, age, balanced diet, regular physical activity during lockdown and PNPLA3 genotype. ^1^ NASH present in 70 patients, out of 80 biopsies available. ^2^ Data available in 188 patients (of which 92 did not increase weight, while 96 increased weight after lockdown).

**Table 2 nutrients-14-00556-t002:** Multivariate analysis of factors associated to weight gain in the entire cohort.

Variable	OR	CI 95%	*p* Value
MODEL 1, Analysis adjusted for age, sex, and diet during lockdown			
Age	0.99	0.97–1.01	0.18
Sex male	0.73	0.45–1.18	0.19
MD during lockdown	0.53	0.34–0.82	0.004
MODEL 2, Analysis adjusted for age, sex, and physical activity during lockdown			
Age	0.99	0.97–1.01	0.11
Sex male	0.75	0.47–1.18	0.21
PA during lockdown	0.60	0.37–0.95	0.03
MODEL 3, Analysis adjusted for age, sex, and PNPLA3			
Age	1.01	0.98–1.02	0.98
Sex male	1.11	0.58–2.15	0.75
PNPLA3 GG	2.4	1.08–5.32	0.03
MODEL 4, Analysis adjusted for age, sex, diet during lockdown, and physical activity during lockdown			
Age	0.98	0.97–1.01	0.10
Sex male	0.77	0.47–1.25	0.28
MD during lockdown	0.54	0.34–0.87	0.01
PA during lockdown	0.58	0.35–0.96	0.03
MODEL 5, Analysis adjusted for age, sex, diet during lockdown, physical activity during lockdown, and PNPLA3			
Age	1.00	0.97–1.02	0.83
Sex male	0.86	0.42–1.76	0.68
MD during lockdown	0.59	0.31–1.13	0.11
PA during lockdown	0.68	0.35–1.30	0.24
PNPLA3 GG	2.391	1.02–5.58	0.04

MD, Mediterranean diet. PA, physical activity.

**Table 3 nutrients-14-00556-t003:** Differences in metabolic and hepatic features after lockdown in the entire cohort according to weight gain.

	Patients without Increased Weight *N* = 186			Patients with Increased Weight *N* = 171		
Variable	Before Lockdown	After Lockdown	*p* Value	Before Lockdown	After Lockdown	*p* Value
Lifestyle						
MD, *n* (%)	91 (49)	100 (54)	0.24	94 (55)	72 (42)	0.91
PA, *n* (%)	65 (35)	67 (36)	0.88	60 (35)	43 (25)	0.02
Metabolic Features						
T2DM, *n* (%)	41 (22)	41 (22)	1.00	43 (25)	43 (25)	1.00
Hypertension, *n* (%)	102 (55)	106 (57)	0.08	94 (55)	94 (55)	1.00
Dislipidemia, *n* (%)	97 (52)	100 (54)	0.16	92 (54)	96 (56)	0.05
Glycemia, mg/dL	106 ± 22	104 ± 21	0.13	106 ± 25	112 ± 32	0.002
HDL cholesterol mg/dL	49 ± 13	49 ± 13	0.94	51 ± 14	51 ± 15	0.22
LDL cholesterol mg/dL	103 ± 97	105 ± 35	0.78	100 ± 31	104 ± 34	0.18
Triglycerides mg/dL	122 (93–164)	121 (94–151)	0.07	114 (89–160)	117 (95–162)	0.40
Liver Disease						
Increased ALT, *n* (%)	50 (27)	41 (22)	0.09	36 (21)	51 (30)	0.02
Increased AST, *n* (%)	19 (10)	20 (11)	1.00	15 (9)	20 (12)	0.16
Increased GGT, *n* (%)	63 (34)	58 (31)	1.00	43 (25)	60 (35)	0.01
US steatosis ➢ 0–1, *n* (%) ➢ 2–3, *n* (%)	89 (48) 97 (52)	81 (44) 105 (56)	0.21	70 (41) 101 (59)	53 (31) 118 (69)	0.002
FIB-4	1.31 (0.95–1.73)	1.29 (1.01–1.82)	0.11	1.28 (0.89–1.64)	1.28 (0.95–1.67)	0.13
FIB-4 < 1.3, *n* (%)	78 (42)	80 (43)	1.00	77 (45)	79 (46)	0.32
FIB-4 > 2.67, *n* (%)	6 (3)	9 (5)	0.16	8 (5)	8 (5)	1.00

MD, Mediterranean diet. PA, physical activity. T2DM, type 2 diabetes. HDL, high-density lipoprotein. LDL, low-density lipoprotein. ALT, alanine aminotransferase. AST, aspartate aminotransferase. GGT, gamma-glutamyltransferase. US, ultrasound. FIB-4, fibrosis 4 score.

## Data Availability

The datasets generated and/or analyzed during the current study are not publicly available but are available from the corresponding author upon reasonable request.

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
