# Peer review of "Interaction between Lifestyle Changes and PNPLA3 Genotype in NAFLD Patients during the COVID-19 Lockdown"

_nutrients, 2022, doi:10.3390/nu14030556_

Round 1
Reviewer 1 Report
In the present study, Cinque et al. performed a retrospective analysis in order to determine the effects of lifestyle habits and PNPLA3 during social lockdown on NAFLD progression. The authors have showed in different models that adherence to mediterranean diet (MD) and physical activity (PA) influence weight gain, but PNPLA3 polymorphism remain the only significant variable in a combined model. All together, COVID-19 lockdown negatively affects NAFLD patients and seems to promote worsening. The results are clearly presented and discussion is well performed,
- How many patients in your cohort suffered from a COVID-19 infection? Did the authors consider this infection as a risk factor (e.g. less PA), which has been demonstrated in previous studies?
- The authors claim that COVID-19 lockdown led to less adherence to MD and less PA, but no data are provided. Can the authors provide data on adherence to MD and PA in non-COVID times? According to table 3
- How many patients could decrease their weight during lockdown? Data on weight course should be included in Table 3
- Details on statistical analysis should be included in the method section.
Author Response
In the present study, Cinque et al. performed a retrospective analysis in order to determine the effects of lifestyle habits and PNPLA3 during social lockdown on NAFLD progression. The authors have showed in different models that adherence to mediterranean diet (MD) and physical activity (PA) influence weight gain, but PNPLA3 polymorphism remain the only significant variable in a combined model. All together, COVID-19 lockdown negatively affects NAFLD patients and seems to promote worsening. The results are clearly presented and discussion is well performed,
1. How many patients in your cohort suffered from a COVID-19 infection? Did the authors consider this infection as a risk factor (e.g. less PA), which has been demonstrated in previous studies?
We really thank the Reviewer for his interesting comment. Nevertheless, in our study only 15 patients out 357 subjects were affected by SARS-COV2 infections. This small number is possibly due to the fact that during post lockdown visits were recorded information only about symptomatic (i.e hospitalized and not-hospitalized) cases, whereas asymptomatic subjects usually did not undergo nasal swab, so could have been missed. In addition, data in our study refers to the first wave of infection (i.e march-june 2020) when strict lockdown measures were introduced in Milan and were able to contain the spread of contagious events.
Therefore, given the very small number of NAFLD infected patients, it is not possible to perform a statistical analysis in order to evaluate the association between Covid-19 infection and less adherence to PA.
2. The authors claim that COVID-19 lockdown led to less adherence to MD and less PA, but no data are provided. Can the authors provide data on adherence to MD and PA in non-COVID times? According to table 3
We really apologize with the Reviewer for not reporting this data clearly. We have now re-wrote the sentence accordingly (line 149 “Before the lockdown, in the entire cohort, 186 (56%) patients referred to following a balanced MD, while 128 (37%) referred that they performed regular physical activity (35% aerobic PA and 2% resistance PA).”).
As for table 3, we have now added these data and also added a comment in both the results section (line 207 “adherence to MD remained stable and low in the whole cohort before and after lockdown, irrespectively of body weight change. Conversely, a lower adherence to PA was significantly evident in the subset of patients who gained weight (35% vs 25%, p=0.02), without any modification in those who did not.”) and the discussion (line 262 “Indeed, modification of lifestyle was the main driver of weight gain, since both the reduction of physical activity (significantly reduced after lockdown) and a low adherence to the Mediterranean diet (around 50%) were independent risk factors for it.”.
- How many patients could decrease their weight during lockdown? Data on weight course should be included in Table 3
We completely agree with the Reviewer, so that data about weight course have now been added in paragraph 3.3 of results (line 168, “Conversely, in 17% of patients body weight remained stable, whereas 35% of the cohort lost weight, with a mean weight loss of 2.7±2.7 Kg.”). Given the fact that table 3 compares information before and after lockdown according to weight gain, we think it would be clearer for the reader to leave information about weight change (i.e weight loss, stable weight or weight gain) only in the results section.
- Details on statistical analysis should be included in the method section.
We really apologize for this mistake. Indeed, it has been a type error during manuscript upload. Now the whole paragraph related to the statistical analysis has been added (line 132, section 2.6 “
Continuous variables were expressed as means ± SD or medians [interquartile range] for normally and not normally distributed variables, respectively. Categorical variables were expressed as absolute and relative frequencies (n, %). In order to compare differences between groups we used the chi-squared test for categorical variables and the unpaired Student’s t-test or the Wilcoxon test for continuous ones, according to data distribution. To evaluate changes in metabolic and lifestyle variables before and after lockdown, a paired t-test and the independent sample proportions were used for continuous and categorical variables. A multivariable logistic regression analyses was performed to evaluate factors associated to increased weight after lockdown, adjusted for confounding factors (age, sex, MD and PA during lock-down and PNPLA3 genotype) and tested, assuming an additive model. A two-tailed p-value ≤ 0.05 was considered statistically significant. Statistical analyses were performed using JMP 15.2 (SAS Institute, Cary, NC).”).
Reviewer 2 Report
- Please make change to the title like: PNPLA3 GG is only important factor the development of NAFLD.
- Authors should ask native English speaker to edit their manuscript.
Author Response
Please make change to the title like: PNPLA3 GG is only important factor the development of NAFLD.
We really thank the Reviewer for its suggestion, as we both estimate the central role of PNPLA3 gene in NAFLD. Nevertheless, as reported in the study, the genetic data was available only in 188 subjects so that a generalization of results by inserting this assertion in the title could be misleading. However, the central role of PNPLA3 should be stressed as for the reviewer suggestion, so a new title could be proposed as "Interaction between lifestyle changes and PNPLA3 genotype in NAFLD patients during the COVID-19 lockdown ".
Authors should ask native English speaker to edit their manuscript.
We agree with the Reviewer and the manuscript has now been fully revised by an English native speaker in order to amend spelling mistakes.
Round 2
Reviewer 1 Report
The authors satisfactorily addressed the issues raised during the review.
Author Response
We really thank the Reviewer for his answer.
Best regards.
Felice Cinque and Rosa Lombardi